# Predictors of in-hospital mortality in stroke patients

**Vindya Shalini Ranasinghe**[1]*, **Manoji Pathirage**[2], **Indika Bandara Gawarammana**[2]

**1** Department of Basic Sciences, Faculty of Allied Health Sciences, University of Peradeniya, Peradeniya, Sri Lanka, **2** Department of Medicine, Faculty of Medicine, University of Peradeniya, Peradeniya, Sri Lanka

* vindya121@gmail.com

## Abstract

In-hospital mortality is a good indicator to assess the efficacy of stroke care. Identifying the predictors of in-hospital mortality is important to advance the stroke outcome and plan the future strategies of stroke management. This was a prospective cohort study conducted at a tertiary referral center in Sri Lanka to identify the possible predictors of in-hospital mortality. The study included 246 confirmed stroke patients. The diagnosis of stroke was established on the clinical history, examination and neuroimaging. The differentiation of stroke in to haemorrhagic type and ischaemic type was based on the results of computed tomography. In all patients, demographic data, comorbidities, clinical signs (pulse rate, respiratory rate, systolic blood pressure, diastolic blood pressure, on admission Glasgow Coma Scale (GCS) score) and imaging findings were recorded. All patients were followed up throughout their hospital course and the in-hospital mortality was recorded. In hospital mortality was defined as the deaths which occurred due to stroke after 24 hours of hospital admission. The incidence of in-hospital mortality was 11.7% (95% confidence interval: 8–16.4). The mean day of in-hospital deaths to occur was 5.9 days (SD ± 3.8 Min 2 Max 20). According to multivariate logistic regression analysis on admission GCS score (Odds Ratio (OR)-0.71) and haemorrhagic stroke type (OR-5.12) predict the in-hospital mortality. The area under the curve of receiver operating curve drawn for the on admission GCS score was 0.78 with a sensitivity of 96.31% and specificity of 41.38% for a patient presented with the GCS score of <10. On admission GCS and haemorrhagic stroke are independent predictors of in-hospital mortality. Thus, a special attention should be given to the patients with low GCS score and haemorrhagic strokes for reducing rates of in-hospital mortality.

## 1. Introduction

Stroke is one of the most devastating neurological condition which results in a huge burden on health and economy. It is a leading cause of death globally which is only second to heart diseases [1]. According to statistics in 2019, there were 6.6 million deaths and 143 million disability adjusted life years (DALYs) lost due to stroke worldwide [2]. The prevalence of stroke is likely to increase globally in years to come as the population is ageing. It is estimated that population over 65 years is increased by 9 million per year globally [3]. This demographic

**Data Availability Statement:** All data generated or analyzed during this study are included in this published article.

**Funding:** This work was supported by University grants of University of Peradeniya, Sri Lanka [Grant

No: URG/2018/08/AHS]. VSR is the PI for this grant. The funders had no role in study design, data collection and analysis, decision to publish or preparation of the manuscript.

**Competing interests:** The authors have declared no competing interests exist.

transition towards an ageing population is observed in Sri Lanka as well and it could result in increased stroke burden [4]. Epidemiological data on stroke has revealed that there is shift of stroke burden from high income countries to low and middle income countries [5]. In fact, 71% of stroke mortality and 78% of DALYs lost were reported from low and middle income countries [6]. South Asian countries contribute most to the global stroke mortality accounting more than 40% of all stroke deaths [7].

Stroke mortality is a good indicator to assess the efficacy of stroke care. In-hospital mortality is even a better indicator as it clearly reflects the efficiency of medical care in managing the disease. Identifying the predictors of in-hospital mortality could definitely advance the stroke outcome [8]. There has been no study done to evaluate the stroke mortality in Sri Lanka. A review done on stroke epidemiology in South Asian countries has also shown that data on stroke mortality and associated risk factors of stroke are scarce in Sri Lanka [7]. Therefore our aim of this study is to evaluate the incidence and predictors of in-hospital mortality in stroke as that information is important in planning future strategies in stroke management.

## 2. Methodology

### 2.1 Design and setting

A prospective cohort study was carried out on 246 stroke patients admitted to Teaching Hospital Peradeniya located in the central province of Sri Lanka. It is a tertiary care center which receive both direct admissions and transfers from other hospitals in central province of Sri Lanka. There is no specialized center for stroke management and all the patients are managed in general medical wards. Ethical clearance to this study was obtained from the Ethics Review Committee, Faculty of Medicine, University of Peradeniya, Sri Lanka (2018/EC/44). Written informed consent was obtained from the patient or, in instances where the patient was not clinically fit to do so, from the next of kin.

### 2.2 Data collection

The current paper is based on the data from patients of stroke recruited prospectively from 1st of June 2019 to 12th of February 2021. The eligibility for the study was considered according to the World Health Organization definition of the stroke and it is the sudden deterioration of brain function due to interrupted blood supply which lasts more than 24 hours [9]. The diagnosis of stroke was established on the clinical history, examination and neuroimaging. The differentiation of stroke in to haemorrhagic type and ischaemic type was based on the results of computed tomography. In all patients, demographic data, comorbidities, clinical signs (pulse rate, respiratory rate, systolic blood pressure, diastolic blood pressure, Glasgow Coma Scale (GCS) score) and imaging findings were recorded. Serum electrolyte test was performed in all stroke patients and hyponatremia was defined as serum $Na^+$ less than 131mmol/l [10]. All patients were followed up throughout their hospital course and the in-hospital mortality was recorded. In hospital mortality was defined as the deaths which occurred due to stroke after 24 hours of hospital admission.

### 2.3 Statistical analysis

Fisher's exact test or Pearson chi square test for categorical variables and student T test or Mann Whitney U test for continuous variables were used to compare the characteristics between in hospital deceased and in hospital alive groups. Forward multivariate logistic regression model was used to evaluate the predictors of in-hospital mortality. Receiver operating characteristic (ROC) curve was drawn to the predictor which showed significant P value in

**Table 1. Age and gender distribution of patients with stroke.**

| Age categories | Gender | | | | | |
|---|---|---|---|---|---|---|
| | Male | % | Female | % | Total | % |
| <20 | 0 | 0 | 1 | 0.4 | 1 | 0.4 |
| 20–39 | 3 | 1.2 | 1 | 0.4 | 4 | 1.6 |
| 40–59 | 26 | 10.6 | 30 | 12.2 | 56 | 22.8 |
| 60–79 | 70 | 28.5 | 71 | 28.8 | 141 | 57.3 |
| > = 80 | 21 | 8.5 | 23 | 9.4 | 44 | 17.9 |
| Total | 120 | 48.8 | 126 | 51.2 | 246 | 100 |

multivariate analysis in order to evaluate more on the predictive ability of that factor. A P value of 0.05 or less is considered to be significant. Statistical analysis was performed by using STATA Version 16 (StataCorp. 2019. *Stata Statistical Software*: College Station, TX: StataCorp LLC.).

## 3. Results

### 3.1 Study participant characteristics

The study sample comprised of 246 proven cases of stroke patients. The mean age of the study participants was 68.14 years (SD ± 12.71). The percentage of males was 48.7% and the mean age was 68.25 years (SD ± 12.25). The mean age of the females was 67.8 years (SD ± 12.65) and there was no statistical difference between the mean age of males and females (P = 0.778). Most of the patients belonged to the age group of 60–79 years (Table 1). Among the study group, 27.2% were smokers and 35.7% were alcohol abusers.

In the study sample, 56.09% patients were on treatment for hypertension. However there was 21.1% of patients who were not diagnosed previously, but having high blood pressure on admission and 28.8% of hypertensive patients had uncontrolled hypertension while on treatment. Diabetes mellitus comprised of 30.8% of patients and 13% of patients were on lipid lowering drugs.

Of the 246 stroke patients, 79.7% had ischaemic stroke and 20.3% had haemorrhagic stroke. In the study sample the prevalence of left sided stroke was 39.02%, 41.8% had right sided stroke and 19.1% had bilateral changes. The stroke type and gender distribution is given in the Table 2.

### 3.2 In-hospital mortality evaluation

Out of 246 stroke patients, 29 (11.7%) patients died during the hospital stay (95% confidence interval (CI): 8–16.4). The mean day of in-hospital deaths to occur was 5.9 days (SD ± 3.8 Min 2 Max 20). The average length of the hospital stay was 3.2 days after admitting for stroke. There was a statistical difference between the stroke type and on admission GCS score between the in-hospital deceased group and in-hospital alive group (Table 3).

**Table 2. Stroke type and gender distribution of patients with stroke.**

| Stroke type | Gender | | | | | |
|---|---|---|---|---|---|---|
| | Male | % | Female | % | Total | % |
| Ischaemic | 93 | 37.8 | 103 | 41.9 | 196 | 79.7 |
| Haemorrhagic | 27 | 10.9 | 23 | 9.4 | 50 | 20.3 |
| Total | 120 | 48.7 | 126 | 51.3 | 246 | 100 |

**Table 3. Comparison of characteristics between the in-hospital deceased group and in hospital alive group.**

| Stroke Patients (*n*-246) | Deceased (In hospital) (*n*-29) | Alive (*n*-217) | P |
|---|---|---|---|
| Age | 71.34 (SD±12.58) | 67.58 (SD±12.62) | 0.08 |
| Gender | | | |
| Male | 14 (5.69%) | 106 (43.09%) | 1.00 |
| Female | 15 (6.1%) | 111 (45.12%) | |
| Alcohol status | 9 (3.66%) | 79 (32.11%) | 0.68 |
| Smoking status | 8 (3.25%) | 59 (23.98%) | 1.00 |
| Hypertension | 18 (7.32%) | 121 (49.19%) | 0.56 |
| Diabetes mellitus | 5 (2.03%) | 71 (28.86%) | 0.13 |
| Dyslipidemia | 1 (0.41%) | 31 (12.6%) | 0.14 |
| On admission GCS | 10.68 (SD±3.41) | 13.97 (SD±1.98) | **0.00** |
| Systolic blood pressure mmHg | 169.58 (SD±49.03) | 156.3 (SD±30.58) | 0.09 |
| Diastolic blood pressure mmHg | 95.68 (SD±26.21) | 90.46 (SD±16.4) | 0.14 |
| Pulse rate | 79.8 (SD±12.34) | 78.8 (SD±9.59) | 0.81 |
| Respiratory rate | 17.6 (SD±3.45) | 17.9 (SD±3.08) | 0.58 |
| Stroke type | | | |
| Ischaemic | 13 (5.28%) | 183 (74.39%) | **0.00** |
| Haemorrhagic | 16 (6.5%) | 34 (13.82%) | |
| Side of the stroke | | | |
| Left | 10 (4.07%) | 86 (34.96%) | |
| Right | 13 (5.28%) | 90 (36.59%) | 0.87 |
| B/L | 6 (2.44%) | 41 (16.67%) | |
| Development of hyponatremia | 8 (3.25%) | 39 (15.85%) | 0.22 |

[1]Fisher's exact test or Pearson chi square test and student t-test or Mann Whitney u test were performed to obtain the results.

[2]Bold indicates statistical significance.

## 3.3 Predictors of in-hospital mortality

Forward Multivariate logistic regression model which illustrates the predictors of in-hospital mortality is given in the Table 4. The overall model was significant (P = 0.000, pseudo $R^2$ = 0.29 with a log likelihood of -63.38).

**Table 4. Multivariate: Logistic regression model for predictors of in-hospital mortality.**

| | Odds Ratio | 95% confidence interval | | P |
|---|---|---|---|---|
| Age | 1.02 | 0.98 | 1.06 | 0.38 |
| Gender | 1.08 | 0.41 | 2.83 | 0.870 |
| Female | | | | |
| Hypertension | 1.74 | 0.64 | 4.79 | 0.640 |
| Diabetes mellitus | 0.31 | 0.09 | 1.10 | 0.070 |
| Systolic Blood pressure | 1 | 0.98 | 1.02 | 0.960 |
| Diastolic blood pressure | 1 | 0.96 | 1.04 | 0.960 |
| On admission GCS | **0.71** | **0.60** | **0.83** | **0.000** |
| Stroke type Haemorrhagic | **5.12** | **1.82** | **14.38** | **0.002** |
| Presence of hyponatremia | 0.99 | 0.33 | 3.02 | 0.990 |
| Development of Aspiration pneumonia | 1.55 | 0.46 | 5.2 | 0.480 |

Foot note: Bold indicates statistical significance.

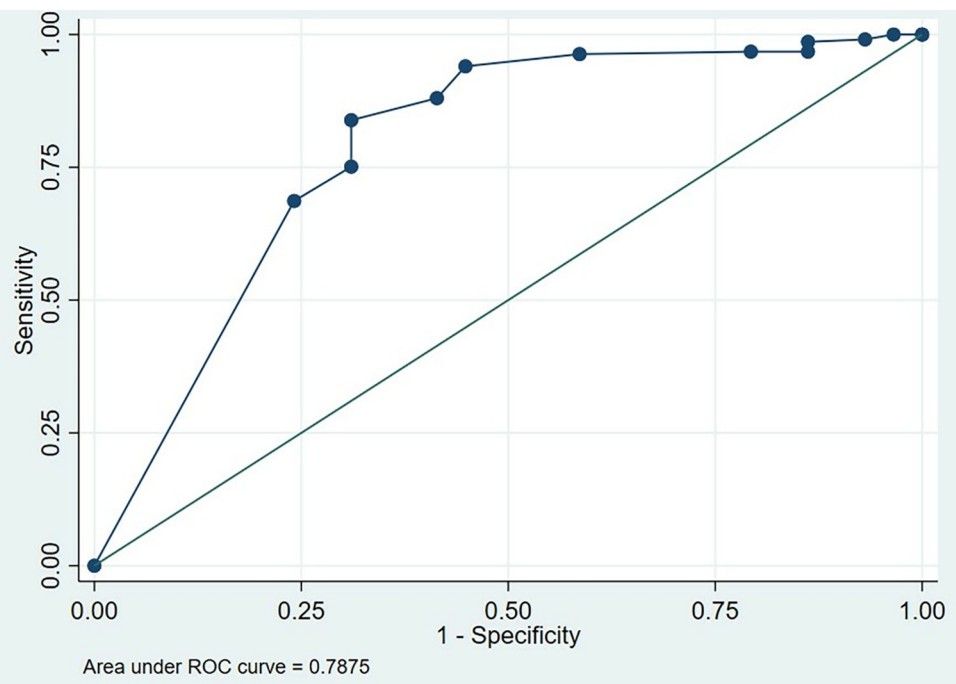

**Fig 1. ROC for on admission GCS score and in-hospital mortality.** Receiver operating characteristic curve (ROC) for prediction of in-hospital mortality based on the on admission Glasgow Coma Scale (GCS) score. Area under the ROC curve was 0.78 for the GCS score less than 10.

The area under the curve (AUC) in the ROC drawn for on admission GCS score was 0.78 with a sensitivity of 96.31% and specificity of 41.38% for a GCS score <10 (Fig 1).

## 4. Discussion

The study sample comprised of 246 diagnosed stroke patients. The majority of patients (79.7%) presented with ischaemic stroke. The incidence of in-hospital mortality was 11.7% and the mean day of in-hospital death to occur was 5.9 days. According to multivariate logistic regression model, haemorrhagic stroke type and on admission GCS score were the independent predictors of in-hospital mortality. Furthermore, the ROC analysis revealed that GCS score <10 was having a moderate predictive ability in predicting in-hospital mortality.

The incidence of in-hospital mortality was 11.7% in our study group. One Indian study has reported a higher incidence of in-hospital mortality which is 26% [11]. One Pakistan study has reported an incidence of 9% [12]. The difference of these figures could be due to different ways of diagnosing the stroke, treatment approaches and stroke care given [1].

One study done in Nepal has concluded that lower GCS score is a predictor of in-hospital mortality [13]. This finding is consistent with our study as the AUC of ROC for GCS score was 0.78 which showed moderate predictive ability of GCS score in predicting in-hospital morality. The prediction could be done with a sensitivity of 96.31% and specificity of 41.38% for a patient who presented with a GCS score <10. One Nigerian study also concluded that GCS score <10 is an independent predictor of in-hospital mortality [14]. Another Nigerian study has revealed a much lower GCS score which is 8 as the cutoff point when predicting in–hospital mortality [15].

Previous literature has suggested that haemorrhagic stroke is responsible for higher number of in-hospital fatalities [8]. Fekadu et al. also pointed the fact that haemorrhagic stroke is

accountable for the most of in-hospital mortality and early mortality in stroke patients [1]. Our study findings also suggest that haemorrhagic stroke is an independent predictor of in-hospital mortality with a significantly high odds ratio of 5.12. An African study done on 325 stroke patients has also revealed that haemorrhagic stroke is an independent predictor of in-hospital mortality with a hazard ratio of 5.65 and P value of 0.0003 [16].

According to our study results, hyponatremia is not an independent predictor of in-hospital mortality. However a study done on 464 intracerebral haemorrhage (ICH) patients has concluded that hyponatremia is an independent predictor of in hospital mortality in ICH patients [17].

## 5. Conclusion

This is the first population based study done on predictors of in-hospital mortality in Sri Lanka. Patients with GCS score<10 and haemorrhagic stroke sub type are more susceptible to die during the hospital stay. Thus, a special attention should be given to the patients with low GCS score and haemorrhagic strokes for reducing the rates of in-hospital mortality.

## Acknowledgments

The authors wish to thank the University of Peradeniya for providing funds and the staff of the medical wards, Teaching Hospital Peradeniya for their support in data collection process.

## Author Contributions

**Conceptualization:** Vindya Shalini Ranasinghe, Manoji Pathirage, Indika Bandara Gawarammana.

**Data curation:** Vindya Shalini Ranasinghe, Manoji Pathirage, Indika Bandara Gawarammana.

**Formal analysis:** Vindya Shalini Ranasinghe.

**Funding acquisition:** Vindya Shalini Ranasinghe, Indika Bandara Gawarammana.

**Investigation:** Vindya Shalini Ranasinghe, Indika Bandara Gawarammana.

**Methodology:** Vindya Shalini Ranasinghe, Manoji Pathirage, Indika Bandara Gawarammana.

**Project administration:** Vindya Shalini Ranasinghe.

**Resources:** Vindya Shalini Ranasinghe.

**Software:** Vindya Shalini Ranasinghe.

**Supervision:** Manoji Pathirage, Indika Bandara Gawarammana.

**Writing – original draft:** Vindya Shalini Ranasinghe.

**Writing – review & editing:** Manoji Pathirage, Indika Bandara Gawarammana.

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
