## [Decision Letter · Decision Letter 0]

23 Nov 2022

PGPH-D-22-01678

Title of the article: Predictors of In-hospital Mortality in Stroke Patients

Dear Ranasinghe,

Thank you for submitting your manuscript to PLOS Global Public Health. After careful consideration, we feel that it has merit but does not fully meet PLOS Global Public Health’s publication criteria as it currently stands. Therefore, we invite you to submit a revised version of the manuscript that addresses the points raised during the review process.

We look forward to receiving your revised manuscript.

Kind regards,

Collins Otieno Asweto, PhD

Academic Editor

Journal Requirements:

2. Please provide separate figure files in .tif or .eps format.

Additional Editor Comments (if provided):

Reviewers' comments:

Reviewer's Responses to Questions

**Comments to the Author**

1. Does this manuscript meet PLOS Global Public Health’s publication criteria? Is the manuscript technically sound, and do the data support the conclusions? The manuscript must describe methodologically and ethically rigorous research with conclusions that are appropriately drawn based on the data presented.

Reviewer #1: Yes

Reviewer #2: Yes

2. Has the statistical analysis been performed appropriately and rigorously?

Reviewer #1: Yes

Reviewer #2: Yes

3. Have the authors made all data underlying the findings in their manuscript fully available (please refer to the Data Availability Statement at the start of the manuscript PDF file)?

Reviewer #1: Yes

Reviewer #2: Yes

4. Is the manuscript presented in an intelligible fashion and written in standard English?

Reviewer #1: Yes

Reviewer #2: Yes

5. Review Comments to the Author

Reviewer #1: This is a hospital-based study conducted in Sri Lanka that focuses on identifying potential predictors of in-hospital mortality in stroke patients. The study comprises a cohort of 246 patients who were admitted to a facility in the country. Clinically and radiologically confirmed stroke patients were prospectively evaluated throughout their stay at the hospital. After considering various factors, the study identified patients with glasgow coma scale (GCS) score < 10 and hemorrhagic stroke type are at higher disk of mortality compared to others. As this population-based study is one of the first in Sri Lanka, it may serve as a platform for clinical researchers and physicians to further understand the management of stroke patients. The study will further help to conduct similar research in other parts of the country. This study is based on 246 patients, so it is important to conduct similar research on a larger cohort.

As the researchers have stated that specialty stroke centers are unavailable in the region, the patient care approach differs compared to western healthcare facilities. Therefore, this study is limited to the subcontinent, which has similar healthcare settings. Also, the researchers have not published the data about the length of the stay of the patients considered in this study at the hospital. It will be helpful to know how long on an average patient stayed in the hospital after admitting for stroke.

In "Introduction"- in the third line, statistics are included of 2013, it would be more appropriate if recent statistics are presented.

Reviewer #2: General; Clear and simple language was used

Background; concise and enables the reader to understand what is intended to be done. The authors however did not show whether stroke is a big concern in Sri Lanka

Methods; Detailed and clear

Results; some of the row percentages do no add up to 100

Discussion; repetition of results in the first paragraph

6. PLOS authors have the option to publish the peer review history of their article (what does this mean?). If published, this will include your full peer review and any attached files.

**Do you want your identity to be public for this peer review?** For information about this choice, including consent withdrawal, please see our Privacy Policy.

Reviewer #1: No

Reviewer #2: No

---

## [Decision Letter · Decision Letter 1]

3 Jan 2023

Title of the article: Predictors of In-hospital Mortality in Stroke Patients

PGPH-D-22-01678R1

Dear Ranasinghe,

We are pleased to inform you that your manuscript 'Title of the article: Predictors of In-hospital Mortality in Stroke Patients' has been provisionally accepted for publication in PLOS Global Public Health.

Best regards,

Collins Otieno Asweto, PhD

Academic Editor

PLOS Global Public HealthReviewer Comments (if any, and for reference):

Reviewer's Responses to Questions

**Comments to the Author**

1. If the authors have adequately addressed your comments raised in a previous round of review and you feel that this manuscript is now acceptable for publication, you may indicate that here to bypass the “Comments to the Author” section, enter your conflict of interest statement in the “Confidential to Editor” section, and submit your "Accept" recommendation.

Reviewer #1: All comments have been addressed

Reviewer #2: All comments have been addressed

2. Does this manuscript meet PLOS Global Public Health’s publication criteria? Is the manuscript technically sound, and do the data support the conclusions? The manuscript must describe methodologically and ethically rigorous research with conclusions that are appropriately drawn based on the data presented.

Reviewer #1: Yes

Reviewer #2: Yes

3. Has the statistical analysis been performed appropriately and rigorously?

Reviewer #1: Yes

Reviewer #2: Yes

4. Have the authors made all data underlying the findings in their manuscript fully available (please refer to the Data Availability Statement at the start of the manuscript PDF file)?

Reviewer #1: Yes

Reviewer #2: Yes

5. Is the manuscript presented in an intelligible fashion and written in standard English?

Reviewer #1: Yes

Reviewer #2: Yes

6. Review Comments to the Author

Reviewer #1: Thank you for addressing the suggestions.

Reviewer #2: Authors have responded well to the comments raised in the previous round of review

One comment was partially addressed; Indicating the burden of stroke mortality in the introduction section

They can consider referencing the following/similar articles for this purpose; https://www.karger.com/Article/FullText/515890

7. PLOS authors have the option to publish the peer review history of their article (what does this mean?). If published, this will include your full peer review and any attached files.

**Do you want your identity to be public for this peer review?** For information about this choice, including consent withdrawal, please see our Privacy Policy.

Reviewer #1: No

Reviewer #2: No
